# Cohort study of high-intensity focused ultrasound in the treatment of localised prostate cancer treatment: Medium-term results from a single centre

Yen-Ting Wu[1‡], Po Hui Chiang[1,2‡]*

1 Department of Urology, Kaohsiung Chang Gung Memorial Hospital and Chang Gung University College of Medicine, Kaohsiung, Taiwan, 2 College of Medicine, Kaohsiung Medical University, Kaohsiung, Taiwan

‡ These authors share first authorship on this work.
* cphtem@yahoo.com.tw

**Data Availability Statement:** All relevant data are within the paper.

**Funding:** The authors received no specific funding for this work.

## Abstract

We report medium-term results in men receiving primary whole-gland HIFU (WG-HIFU) and following salvage treatment. One hundred and twenty-eight patients in a single hospital were enrolled. The enrolled patients were treated with WG-HIFU for primary localized prostate cancer. Salvage treatment include androgen deprivation therapy, secondary HIFU and salvage radiation therapy. Our primary outcomes were biochemical recurrence–free survival, salvage treatment–free survival, and metastasis-free survival. Secondary outcomes included urinary incontinence, de novo erectile dysfunction, acute epididymitis, bladder neck contracture, and urethral stricture. The 5-year biochemical recurrence–free survival rates were 85.7%, 82.7%, and 45.2% for D'Amico low-, intermediate-, and high-risk groups, respectively. Multivariate analysis revealed high risk group is the only predictor of significant shorter biochemical recurrence free survival, salvage treatment free survival, and metastasis free survival. Of 38 patients receiving salvage treatment after biochemical recurrence, 29 (76.3%) became free from biochemical recurrence. Rates of the adverse events of urinary incontinence, acute epididymitis, bladder neck contracture or urethral stricture, and de novo erectile dysfunction were 2.3%, 10.9%, 20.3%, 65.6%, respectively. In conclusion, WG-HIFU is an effective treatment option for localised prostate cancer, especially in D'Amico low- and intermediate-risk cases. The success rate of salvage treatment with radiation therapy and secondary HIFU for biochemical recurrence was acceptable. Fewer adverse events were caused by HIFU, especially incontinence and erectile dysfunction, than by radical prostatectomy and radiotherapy.

## Introduction

The incidence of prostate cancer in Taiwan, although lower than that in Western countries, has been increasing. According to a Taiwanese annual cancer report in 2016, prostate cancer

**Competing interests:** The authors have declared that no competing interests exist.

was the fifth most commonly diagnosed cancer in Taiwan, with a median age at diagnosis of 73 years old [1]. Thus, prostate cancer is becoming a serious health problem, especially in an ageing society. The established and definitive treatment for localised prostate cancer includes radical prostatectomy and radiotherapy; data on long-term outcomes of both treatments are available. However, many patients are concerned about periprocedural adverse events [2, 3], and aged patients, who tend to have comorbidities, have higher perioperative risk [4]; such patients may not tolerate major surgery and tend to be afraid of adverse events from radiotherapy.

Minimally invasive treatment modalities such as cryosurgical ablation of prostate and high-intensity focused ultrasound (HIFU) have been developed for patients with localised prostate cancer [5, 6]. HIFU uses a focused ultrasound wave that mechanically and thermally induces tissue damage, which causes coagulative necrosis through tissue cavitation and temperature elevation [7]. Long-term prospective comparative data on oncological outcomes after primary WG-HIFU are scarce [8]. Tsakiris et al. recommended HIFU, determining it to be oncologically safe for patients with stage T1c to T3 prostate cancer [9]. Data on outcomes of salvage treatment after primary HIFU for localised prostate cancer are also scarce. As HIFU devices evolve, the reported rate of adverse events has been reduced and is now at an acceptable level [6]. In this paper, we present our findings on the functional and medium-term oncological outcomes from our cohort study of men with localised prostate cancer who were treated with primary WG-HIFU. Outcomes of salvage treatment with radiotherapy and secondary HIFU are also presented.

## Materials and methods

This retrospective single-institute study was approved by Kaohsiung Chang Gung Memorial Hospital Institutional Review Board (IRB number: 201101264B0). The IRB waived the requirement for informed consent. At our institution, 405 patients with prostate cancer have been treated with HIFU between December 2009 and July 2019. All patients were treated using Ablatherm® Integrated Imaging (EDAP TMS SA, Vaulx-en-Velin, France) with transrectal ultrasonography guidance under general or spinal anaesthesia. From December 2009 to February 2015, 161 patients who were newly diagnosed with prostate cancer were enrolled. Of these patients, 13 patients received HIFU as salvage treatment for advanced prostate cancer. The remaining 148 patients underwent WG-HIFU for localised prostate cancer. Among these patients, 20 patients were excluded because their follow-up durations were less than 30 months. In total, 128 patients were included (Fig 1). All patients underwent either magnetic resonance imaging (MRI) or computed tomography of the pelvis, in addition to a bone scan for preoperative staging. Cancer staging was done according to the American Joint Committee on Cancer (7th edition) prostate cancer staging system. All patients underwent transurethral resection of the prostate 4 weeks before the HIFU procedure (if prostate volume was ≥40 ml) or simultaneously with the HIFU procedure (if prostate volume was <40 ml). Risk of treatment failure was stratified according to D'Amico risk classification into low-, intermediate-, and high-risk groups. Our institutional protocol for follow-up after HIFU is based on the 3-monthly postoperative prostate-specific antigen (PSA) level. Biochemical recurrence, according to the Phoenix definition, is a post-HIFU PSA nadir +2 ng/ml. Post-HIFU prostate biopsy was not routinely arranged. It might be arranged because of biochemical recurrence. It might also be arranged for patients not meeting criteria of biochemical recurrence but worry about continuous elevation of PSA. If biochemical recurrence is detected, salvage treatment is arranged according to prostate biopsy results. In this study population, salvage therapies included secondary HIFU (34.2%) if residual cancer cells were present, in addition to radiation

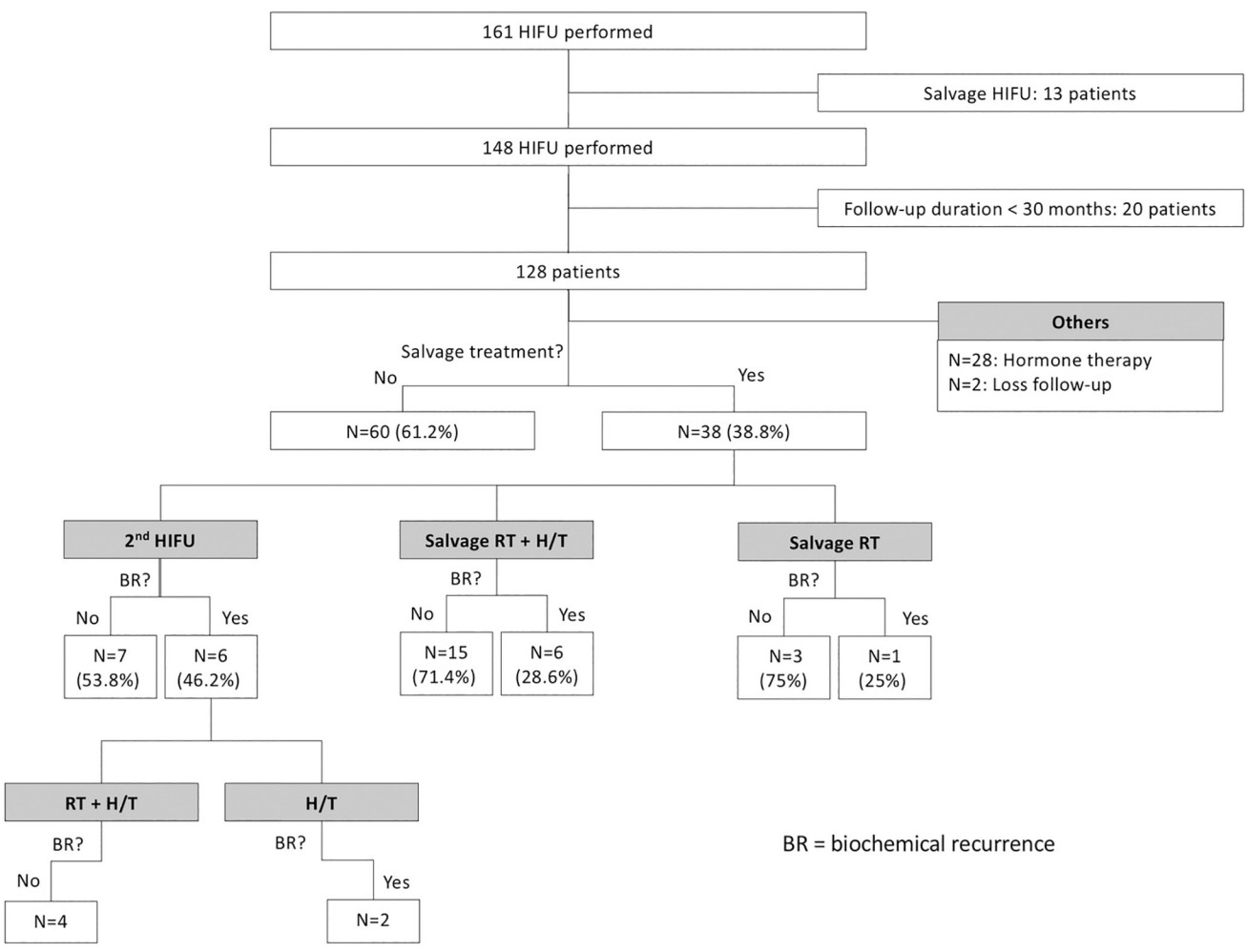

**Fig 1. Outcome after salvage treatment.**

therapy (10.5%) or radiation therapy plus androgen deprivation therapy (ADT; 55.3%). Whether patients receive salvage treatment may be related to the results of prostate biopsy and patient's preference. Of patients with residual tumor noted, secondary HIFU will be arranged. Of patients with local lymph node metastasis suspected, salvage radiotherapy will be suggested. Salvage radiotherapy with half to one year of ADT is the standard strategy. Primary outcomes were biochemical recurrence–free survival (BRFS), salvage treatment–free survival, and metastasis-free survival.

The dose of salvage radiotherapy after HIFU is between 66 and 70 Gy, whereas the dose of primary radiotherapy is usually 74 Gy. The target of salvage radiotherapy includes the prostate and seminal vesicle, but may also include the pelvis lymph node, according to Roach's formula. The standard mode of delivery for primary HIFU is 100% acoustic power with a 6-s pulse of energy to create each discrete HIFU lesion, with a 4-s delay between each shot. The salvage mode of delivery for secondary HIFU is 90% acoustic power with a 4-s pulse and a 6-s waiting period.

The secondary outcomes were urinary incontinence (defined as one or more pads used daily for more than 3 months), *de novo* erectile dysfunction, acute epididymitis, bladder neck

contracture, and urethral stricture (which required surgical treatment under anaesthesia). From March 2015, prophylactic bilateral vasectomy was performed immediately before HIFU and Bougienage at an outpatient clinic during follow-up to prevent postoperative epididymitis and urethral stricture, respectively.

MedCalc software (version 18.9.1) was used for all statistical analyses. Chi-square tests and two-sample *t* tests were used for intergroup comparisons, and the Kaplan–Meier test was used for time-to-event analysis. A *p* value <0.05 was defined as statistically significant.

## Results

### Baseline demographics

This study included 128 patients. Patient characteristics are summarised in Table 1. Mean age was 68.5 (range: 50.9–88.2). Mean prostate volume was 23.2 ml. According to D'Amico classification, the numbers of patients with low-, intermediate-, and high-risk disease were 14 (11.0%), 52 (40.6%), and 62 (48.4%). Median follow-up duration was 53.7 months (interquartile range [IQR]: 44.0–66.0).

### Oncological outcomes

Overall 5-year biochemical recurrence–free and salvage treatment–free survival rates after primary HIFU were 64.8% and 50.8%, respectively (Table 2). The 5-year biochemical recurrence–free survival rates were 85.7%, 82.7%, and 45.2% for D'Amico low-, intermediate-, and high-risk groups, respectively (Table 2). The 5-year salvage treatment–free survival rates were 71.4%, 69.2%, and 30.6% for D'Amico low-, intermediate-, and high-risk groups, respectively (Table 2). Kaplan–Meier curves revealed significant differences in both biochemical recurrence–free and salvage treatment–free survival rate between different risk groups (Figs 2 and 3). As shown in Table 3, using Cox regression multivariate analysis, high-risk group is

**Table 1. Patient characteristics.**

| Characteristics | |
|---|---|
| Total number of men | 128 |
| Age (yr), mean (range) | 68.5 (50.9–88.2) |
| Prostate volume (ml), mean (range) | 23.2 (6.7–71.1) |
| Gleason score, N (%) | |
| <7 | 35 (27.3%) |
| = 7 | 59 (46.1%) |
| >7 | 34 (26.6%) |
| iPSA, N (%) | |
| <10 | 59 (46.1%) |
| 10–20 | 43 (33.6%) |
| >20 | 26 (20.3%) |
| Stage | |
| <T2b | 81 (63.3%) |
| T2b | 11 (8.6%) |
| >T2b | 36 (28.1%) |
| D'Amico risk group, N (%) | |
| Low | 14 (11.0%) |
| Intermediate | 52 (40.6%) |
| High | 62 (48.4%) |

**Table 2. Oncological outcomes.**

| Characteristics | |
|---|---|
| Follow-up period (mo), median (IQR) | 53.73 (43.98–66.02) |
| Nadir PSA (ng/ml), median (IQR) | 0.10 (0.02–0.42) |
| Time to PSA nadir (mo), median (IQR) | 2.52 (1.10–3.87) |
| Overall biochemical recurrence free survival at 5 yr | 64.8% |
| Biochemical recurrence free survival at 5 yr by D'Amico risk group | |
| Low | 85.7% |
| Intermediate | 82.7% |
| High | 45.2% |
| Overall salvage treatment-free survival at 5 yr | 50.8% |
| Salvage treatment-free survival at 5 yr by D'Amico risk group | |
| Low | 71.4% |
| Intermediate | 69.2% |
| High | 30.6% |
| Overall metastasis free survival at 5 yr | 94.5% |

significantly associated with shorter biochemical recurrence free survival, salvage treatment free survival, and metastasis free survival. Median time to salvage treatment was 15.3 months after primary HIFU. Metastasis was detected in one patient in the intermediate-risk group and six patients in the high-risk group (Fig 4). Median nadir PSA was 0.10 ng/ml (IQR: 0.02–0.42 ng/ml). Median time to nadir PSA was 2.52 months (IQR: 1.10–3.87 months). We excluded 28 of 128 patients who were undergoing ADT during follow-up (Fig 1). Two patients were lost to follow-up within one year of HIFU. Of the remaining 98 patients, 60 (61.2%) patients had no biochemical recurrence and thus did not require any salvage treatment, and 38 (38.8%) patients received salvage treatment due to biochemical recurrence. The results of post-HIFU biopsy were shown as Tables 4 and 5. In addition, the relationships between post-HIFU PSA and post-HIFU prostate biopsy were shown in Table 6. There were 13, 21, and 4 patients receiving salvage HIFU, salvage radiation therapy with hormone therapy, and salvage radiation therapy without hormone therapy, respectively. Three out of four patients who received salvage radiation therapy were biochemical recurrence free. Fifteen out of 21 patients who received salvage radiation therapy with hormone therapy were biochemical recurrence free. In other words, in patients receiving salvage radiation therapy with or without hormone therapy, more than 70% of them became biochemical recurrence free. Thirteen patients underwent secondary HIFU. Seven patients were biochemical recurrence free. Among the six patients who had biochemical recurrence after secondary HIFU, four patients received salvage radiation therapy with hormone therapy as second-line salvage treatment, and all were biochemical recurrence free. In total, 38 (38.8%) of 98 patients received salvage treatment with radiotherapy or salvage HIFU. Twenty-nine patients (29/38, 76.3%) were biochemical recurrence free.

## Functional outcomes

Functional outcomes are summarised in Table 7. With the same cohort, our previous study reported a 65.6% rate of *de novo* erectile dysfunction after primary HIFU. The IIEF-5 score was 22.10 ± 2.62 preoperatively and 9.36 ± 6.33 at 24 months after HIFU [5]. Three patients had urinary incontinence. Acute epididymitis developed in 10.9% of patients. Mean onset time of acute epididymitis was 22.6 days after operation (range: 8–71 days). Bladder neck contracture and urethral stricture, which require further surgical treatment under anaesthesia,

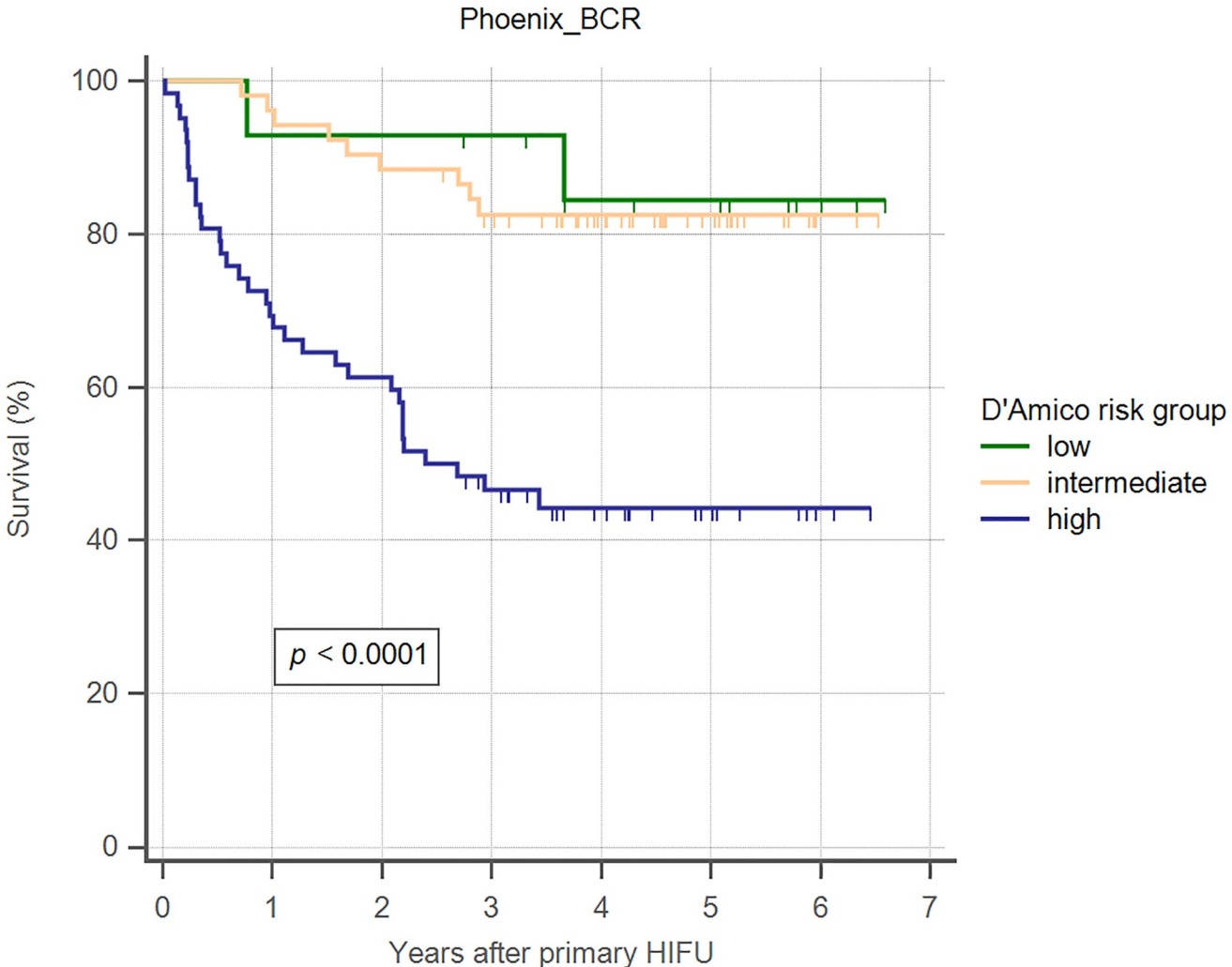

**Fig 2. Kaplan–Meier curves illustrating biochemical recurrence–free survival (Phoenix criteria), by D'Amico risk groups, in men undergoing WG-HIFU for prostate cancer.**

were found in 20.3% of patients. One instance of rectourethral fistula was noted after salvage radiotherapy in our cohort. The rate of adverse events after salvage treatment was acceptable (Table 8).

## Discussion

In Taiwan, the incidence of prostate cancer is increasing during the last decade. becoming a serious health problem. Radical prostatectomy and radiation therapy have been the standard treatments for localised prostate cancer. However, many patients, especially aged patients or patients with multiple comorbidity, may be concerned about adverse events after major surgery and radiation therapy. Minimally invasive treatment modality such as HIFU is a potential alternative option. Sufficient data concerning the long-term oncological outcomes of using HIFU as the treatment of localised prostate cancer remain lacking. Our previous study and another study reported median times to biochemical recurrence of 12.03 and 13.8 months,

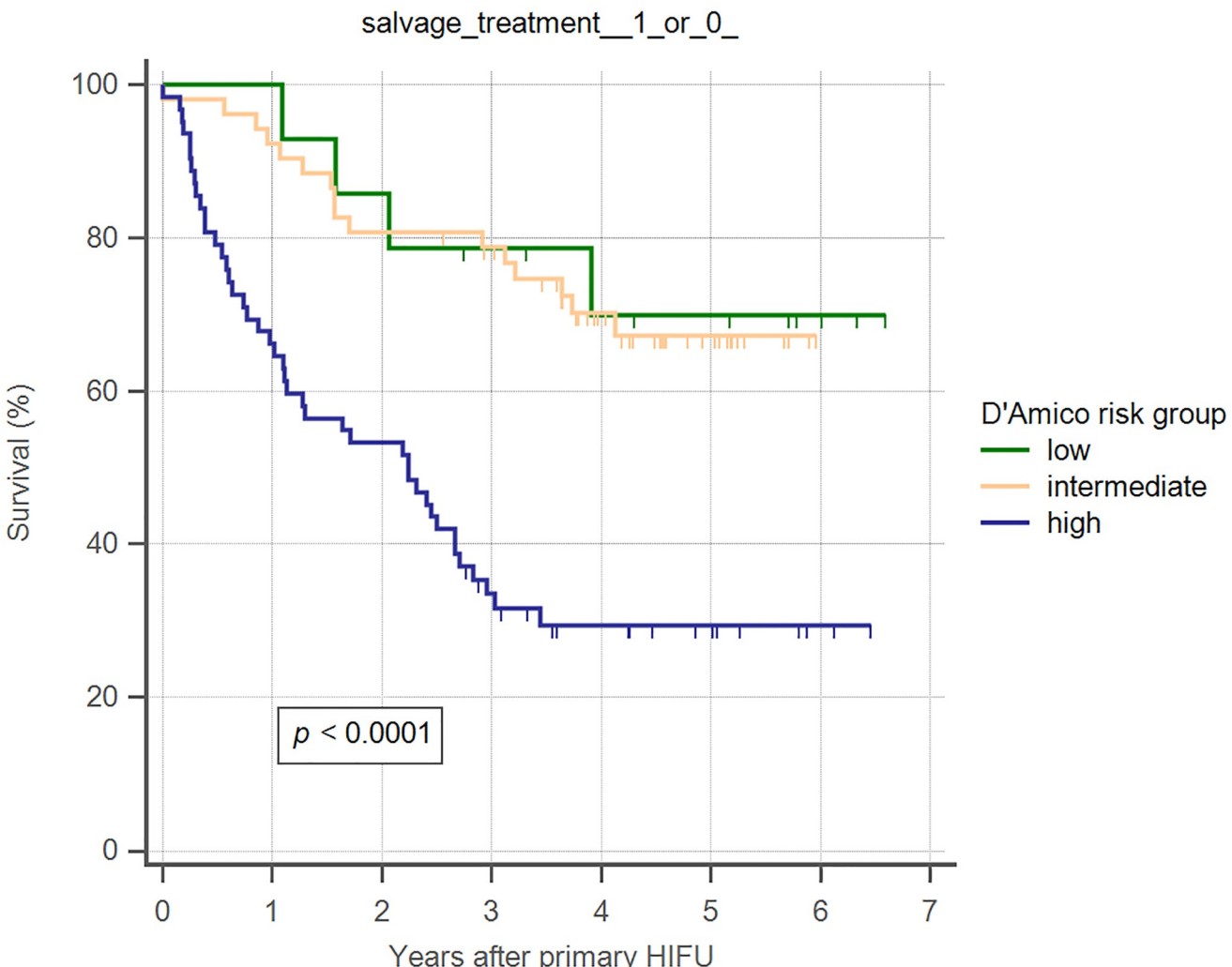

**Fig 3. Kaplan–Meier curves illustrating salvage treatment–free survival, by D'Amico risk groups, in men undergoing WG-HIFU for prostate cancer.**

respectively [10]. Therefore, medium-term data collected after primary HIFU may be used as an indicator of the oncological outcomes of HIFU. Dickinson et al. reported an overall 5-year BRFS rate of 68% in a multicenter cohort where 16% of patients were in the high-risk group [11]. Durán-Rivera et al. also reported a comparable rate of 64.2% after 86.4 months of follow-up in a cohort where 3% were high-risk patients [12]. The proportion of high-risk group patients in our study, at 47%, was larger than those in previous studies, but our observed oncological outcomes are comparable to theirs. In low- and intermediate-risk patients, 5-year salvage treatment–free survival and biochemical recurrence–free survival rates were approximately 70% and 80%, respectively. The overall 5-year BRFS rate after primary HIFU was 64.8%. Median time to nadir PSA was only 2.52 months. Such early biochemical response is consistent with previous studies [13, 14].

There has been no series report on salvage treatment after primary HIFU. Our salvage treatment after primary HIFU included secondary HIFU and radiation therapy. More than 70% of patients who underwent radiation therapy, with or without ADT, were biochemical recurrence

**Table 3. Cox regression multivariate analysis for oncologic outcome after primary WG-HIFU.**

| | Biochemical recurrence free survival | | | |
|---|---|---|---|---|
| Characteristics | Univariate analysis | | Multivariate analysis | |
| | HR (95% CI) | *p* value | HR (95% CI) | *p* value |
| Age | 0.95–1.02 | 0.36 | | |
| Gleason score <7 | ref. | | | |
| Gleason score = 7 | 0.56–2.80 | 0.58 | | |
| Gleason score >7 | 1.29–6.43 | 0.01 | | |
| iPSA <10 | ref. | | | |
| iPSA 10–20 | 0.15–0.66 | 0.002 | | |
| iPSA >20 | 0.14–0.84 | 0.02 | | |
| Stage <T2b | ref. | | | |
| Stage = T2b | 0.66–5.71 | 0.22 | | |
| Stage >T2b | 1.67–5.70 | 0.0003 | | |
| low risk group | ref. | | | |
| intermediate risk group | 0.27–5.75 | 0.78 | | |
| high risk group | 1.29–22.32 | 0.02 | 2.28–8.90 | <0.0001 |
| | Salvage treatment free survival | | | |
| Characteristics | Univariate analysis | | Multivariate analysis | |
| | HR (95% CI) | *p* value | HR (95% CI) | *p* value |
| Age | 0.96–1.03 | 0.75 | | |
| Gleason score <7 | ref. | | | |
| Gleason score = 7 | 0.73–2.64 | 0.31 | | |
| Gleason score >7 | 1.27–4.82 | 0.008 | | |
| iPSA <10 | ref. | | | |
| iPSA 10–20 | 0.23–0.75 | 0.003 | | |
| iPSA >20 | 0.25–0.95 | 0.03 | | |
| Stage <T2b | ref. | | | |
| Stage = T2b | 0.37–2.96 | 0.93 | | |
| Stage >T2b | 1.60–4.38 | 0.0002 | | |
| low risk group | ref. | | | |
| intermediate risk group | 0.44–3.80 | 0.65 | | |
| high risk group | 1.37–10.61 | 0.01 | 1.86–5.21 | <0.0001 |
| | Metastasis free survival | | | |
| Characteristics | Univariate analysis | | Multivariate analysis | |
| | HR (95% CI) | *p* value | HR (95% CI) | *p* value |
| Age | 0.96–1.14 | 0.32 | | |
| Gleason score <7 | ref. | | | |
| Gleason score = 7 | | 0.96 | | |
| Gleason score >7 | | 0.95 | | |
| iPSA <10 | ref. | | | |
| iPSA 10–20 | 0.10–2.67 | 0.43 | | |
| iPSA >20 | | 0.96 | | |
| Stage <T2b | ref. | | | |
| Stage = T2b | | 0.97 | | |
| Stage >T2b | 0.80–16.26 | 0.10 | | |
| low risk group | ref. | | | |
| intermediate risk group | | 0.96 | | |
| high risk group | | 0.95 | 1.03–73.74 | 0.047 |

Abbreviation: ref.: reference

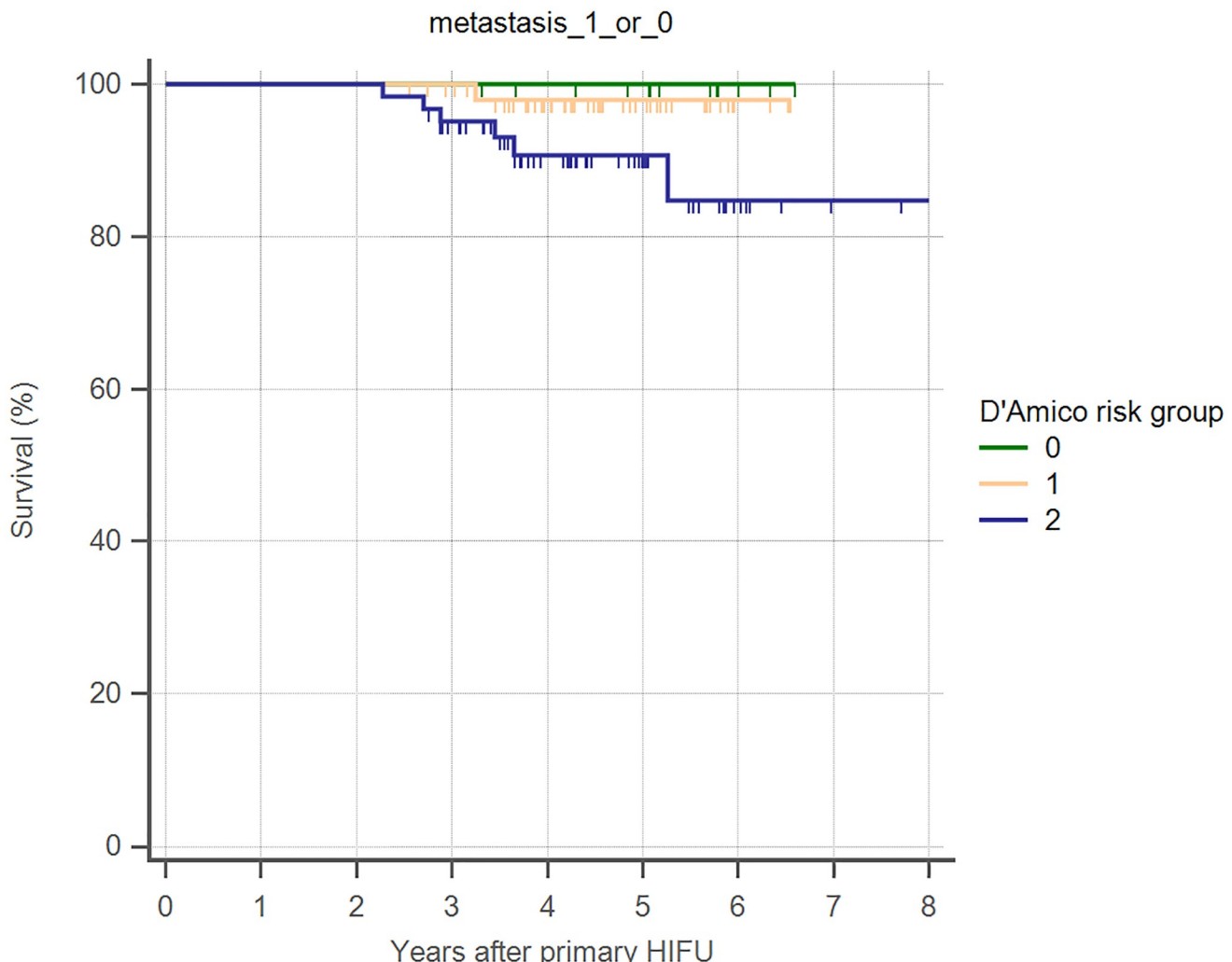

**Fig 4. Kaplan–Meier curves illustrating metastasis-free survival, by D'Amico risk groups, in men undergoing WG-HIFU for prostate cancer.**

free, whereas only approximately half the patients who received secondary HIFU were biochemical recurrence free. In most patients with biochemical recurrence after secondary HIFU, radiation therapy can still be used to achieve BRFS. The overall BRFS rate after primary HIFU, with or without salvage treatment, was 90.6% in our series. Oncological outcomes after primary WG-HIFU are comparable with other treatment modalities for localised prostate cancer. Although the proportion of high-risk patients was disproportionately high in our series, our observed oncological outcomes are comparable to those of other treatment modalities and previous studies of HIFU.

We observed rates of 2.3% for urinary incontinence, 65.6% for *de novo* erectile dysfunction, 11.5% for acute epididymitis, and 20.3% for urethral stricture and bladder neck contracture. These adverse event rates are comparable with those of other studies [15–17]. As evident in Table 3, the rate of urethral stricture and bladder neck contracture was significantly reduced to 10.5% after routine bougienage. Those who received prophylactic bilateral vasectomy tended to have a lower risk of acute epididymitis. No instance of rectourethral fistula was noted in the primary HIFU. However, one rectourethral fistula occurred after salvage radiotherapy.

**Table 4. Results of post-HIFU prostate biopsy.**

| Biopsy methods | No. of patients | Residual cancer | HIFU-biopsy interval (mo), mean |
|---|---|---|---|
| Transurethral biopsy | 32 | 4 | 13.54 |
| Sonography-guided biopsy | 27 | 10 | 15.31 |

According to the EORTC 22991 trial, of 28 patients who received 74 Gy intensity- modulated radiation therapy (IMRT) as primary treatment for localised prostate cancer, 46.4% of patients had grade 1 acute genitourinary toxicity, 25% had grade 2, and 10.7% had grade 3. In the same patient group, 39.3% of patients had grade 1 acute gastrointestinal toxicity and 7.1% had grade 2 [18]. By contrast, in this study, among patients who received salvage radiotherapy (at 66–70 Gy), only 12% had grade 1 acute genitourinary toxicity, and only 8% had grade 1 acute gastrointestinal toxicity. No toxicity grades >1 were reported after salvage radiotherapy. The difference in the incidence of adverse events between patients who received primary and salvage radiotherapy may result from different radiotherapy dosages.

The major limitation of this study was its retrospective design. However, oncological outcomes after primary HIFU for localised prostate cancer were determined to be acceptable relative to radical prostatectomy and radiotherapy. HIFU is an alternative for the treatment of localised prostate cancer with minimal adverse events. Moreover, we identified disease relapse mainly by elevated PSA. However, MRI may provide a more sensitive test. MRI could detect imaging change of residual prostatic tissue where despite not elevating PSA. However, only x/y (%) who meet criteria of biochemical recurrence received post-HIFU MRI due to suspicion of disease relapse.

In general, our study demonstrated a feasible solution for the reduction of postoperative bladder neck contracture, urethral stricture, and acute epididymitis. Our observations of outcomes after salvage treatment with radiotherapy or secondary HIFU were also reported in this

**Table 5. Characteristics of patients receiving post-HIFU prostate biopsy.**

| Characteristics | Residual cancer (N = 14) | Negative (N = 45) |
|---|---|---|
| Age (yr), mean (range) | 67.5 (54.0–82.9) | 69.0 (57.8–85.1) |
| Gleason score, N (%) | | |
| <7 | 2 (14.3%) | 14 (31.1%) |
| = 7 | 10 (71.4%) | 18 (40.0%) |
| >7 | 2 (14.3%) | 13 (28.9%) |
| iPSA, N (%) | | |
| <10 | 8 (57.1%) | 24 (60.0%) |
| 10–20 | 5 (35.7%) | 15 (33.3%) |
| >20 | 1 (7.1%) | 6 (13.3%) |
| Stage (%) | | |
| <T2b | 10 (71.4%) | 27 (60.0%) |
| T2b | 0 (0.0%) | 5 (11.1%) |
| >T2b | 4 (28.6%) | 13 (28.9%) |
| D'Amico risk group, N (%) | | |
| Low | 1 (7.1%) | 8 (17.8%) |
| Intermediate | 7 (50.0%) | 13 (28.9%) |
| High | 6 (42.9%) | 24 (53.3%) |
| Biochemical recurrence, N (%) | 8 (57.1%) | 19 (42.2%) |
| Salvage treatment arranged, N (%) | 14 (100%) | 24 (53.3%) |

**Table 6. Relationships between post-HIFU PSA and post-HIFU prostate biopsy.**

| Sonography-guided biopsy | Positive of malignancy (N = 10) | Negative of malignancy (N = 17) | *p* value |
|---|---|---|---|
| iPSA, mean (ng/ml) | 12.5 | 22.8 | 0.29 |
| post-HIFU PSA nadir, mean (ng/ml) | 0.3 | 1.4 | 0.15 |
| biochemical recurrence, N | 7 | 13 | 0.72 |

**Table 7. Adverse events after primary HIFU.**

| Adverse events | N = 128 |
|---|---|
| Urinary incontinence | 2.3% |
| Acute epididymitis | 10.9% |
| Bladder neck contracture / Urethral stricture | 20.3% |
| De novo erectile dysfunction | 65.6% |

**Table 8. De novo adverse events after salvage treatments.**

| Adverse events | CTCAE v5.0 | Salvage HIFU (N = 13) | Salvage radiotherapy (N = 25) | *p* value |
|---|---|---|---|---|
| Urinary incontinence | Grade 1–2 | 0 (0.0%) | 4 (16.0%) | 0.13 |
| | Grade 3 | 0 (0.0%) | 0 (0.0%) | 0.05 |
| Urinary tract obstruction | Grade 1–2 | 1 (7.7%) | 4 (16.0%) | 0.48 |
| | Grade 3 | 2 (15.4%) | 3 (12.0%) | 0.77 |
| Gastrointestinal disorders | Grade 1–2 | 0 (0.0%) | 5 (20.0%) | 0.09 |
| | Grade 3 | 0 (0.0%) | 1 (4%) | 0.47 |

paper. In total, 90.6% of patients with localised prostate cancer in our institute could achieve BRFS (at a median of 53.7 months) following primary HIFU or subsequent salvage treatment with secondary HIFU or radiotherapy.

## Conclusions

WG-HIFU is an effective treatment option for localised prostate cancer, especially in D'Amico low- and intermediate-risk diseases. Although some patients experienced PSA biochemical recurrence, the success rate of salvage treatment was still acceptable. Salvage radiation therapy may play an important role in biochemical recurrence after primary HIFU. In addition, the complication rate after primary HIFU was lower than that for other treatment modalities. Long-term follow-up of 10 to 15 years is still required for oncological control.

## Acknowledgments

This manuscript was edited by Wallace Academic Editing. We appreciate the Biostatistics Center, Kaohsiung Chang Gung Memorial Hospital.

## Author Contributions

**Conceptualization:** Po Hui Chiang.

**Data curation:** Yen-Ting Wu.

**Formal analysis:** Yen-Ting Wu.

**Investigation:** Yen-Ting Wu, Po Hui Chiang.

**Software:** Yen-Ting Wu.

**Supervision:** Po Hui Chiang.

**Writing – original draft:** Yen-Ting Wu.

**Writing – review & editing:** Po Hui Chiang.

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
