## [Decision Letter · Decision Letter 0]

13 Feb 2020

PONE-D-20-00019

Cohort study of high-intensity focused ultrasound in the treatment of localised prostate cancer treatment: medium-term results from a single centre

PLOS ONE

Dear Professor Chiang,

Thank you for submitting your manuscript to PLOS ONE. After careful consideration, we feel that it has merit but does not fully meet PLOS ONE’s publication criteria as it currently stands. Therefore, we invite you to submit a revised version of the manuscript that addresses the points raised during the review process.

ACADEMIC EDITOR: 

The reviewers raised some questions. Please respond to them in detail. Also, please clearly describe the meaning of "NA" in table 3.

We would appreciate receiving your revised manuscript by Mar 29 2020 11:59PM. To enhance the reproducibility of your results, we recommend that if applicable you deposit your laboratory protocols in protocols.io, where a protocol can be assigned its own identifier (DOI) such that it can be cited independently in the future. For instructions see: http://journals.plos.org/plosone/s/submission-guidelines#loc-laboratory-protocols

We look forward to receiving your revised manuscript.

Kind regards,

Jason Chia-Hsun Hsieh, M.D. Ph.D

Academic Editor

PLOS ONE

Journal Requirements:

2. In ethics statement in the manuscript and in the online submission form, please provide additional information about the patient records used in your retrospective study. Specifically, please ensure that you have discussed whether all data were fully anonymized before you accessed them and/or whether the IRB or ethics committee waived the requirement for informed consent. If patients provided informed written consent to have data from their medical records used in research, please include this information.

a)    Please provide an amended Funding Statement that declares *all* the funding or sources of support received during this specific study (whether external or internal to your organization) as detailed online in our guide for authors at http://journals.plos.org/plosone/s/submit-now.  

b)    Please state what role the funders took in the study.  If any authors received a salary from any of your funders, please state which authors and which funder. If the funders had no role, please state: "The funders had no role in study design, data collection and analysis, decision to publish, or preparation of the manuscript."

Additional Editor Comments (if provided):

The reviewers raised some questions. Please respond to them in detail. Also, please clearly describe the meaning of "NA" in table 3.

Reviewers' comments:

Reviewer's Responses to Questions

**Comments to the Author**

1. Is the manuscript technically sound, and do the data support the conclusions?

Reviewer #1: Partly

Reviewer #2: Partly

2. Has the statistical analysis been performed appropriately and rigorously? 

Reviewer #1: Yes

Reviewer #2: N/A

3. Have the authors made all data underlying the findings in their manuscript fully available?

Reviewer #1: Yes

Reviewer #2: No

4. Is the manuscript presented in an intelligible fashion and written in standard English?

Reviewer #1: Yes

Reviewer #2: Yes

5. Review Comments to the Author

Reviewer #1: The authors provided medium term oncological outcomes of a single institution cohort of 128 men receiving whole gland HIFU for clinically localized prostate cancer.

Major comments:

1. I would specify that the study describes whole gland HIFU. Modify and introduce an abbreviation for whole gland HIFU throughout the manuscript.

2. Abstract does not read well. For example, the first proposition does not fit neither with the background nor with the objective of the study. No information are given regarding the efficacy of salvage treatments. Please address this point.

3. Did all patients received a post whole gland HIFU biopsy? Please clarify

4. A table (or a Kaplan meier in case the follow up biopsy protocol was not standardized) showing the results of the follow up biopsy would be useful.

5. Which were the criteria, if there were any, to choose the salvage treatment strategy? Please describe.

6. If the authors are keen on reporting data regarding the efficacy of salvage treatments is necessary to report also the outcomes of the population receiving any salvage treatment (as subgroup analysis and figures as supplementary)

7. Since all men received a pre mpMRI, isit possible to add any information regarding the presence of visible lesions, PI-RADS score etc?

8. Regarding the functional outcomes, which represent secondary outcomes according to the study’s methods, only data regarding the study population should be reported. All information regarding not included cohort should be removed since not relevant.

9. Unless preop IIEF is available, any data regarding post op IIEF should be removed since not reliable.

10. Adverse events should be put in one single table and, if possible, compared for each procedure (provide a p value if possible)

11. The first sentence of the discussion part is out of context and biased. Please change the beginning of this paragraph

Reviewer #2: 1. The major concern on this works is that it is supported the manuscript a small number of patients and a very short period in this manuscript.

2. Abstract-too much space is spent on descriptive data and not enough space on the main result of the study.

3. The author mentions in materials and methods that 405 patients with prostate cancer have been treated with HIFU from December 2009 to February 2015. From December 2009 to February 2015, 161 patients who were newly diagnosed with prostate cancer were enrolled. In total, 128 patients were included (Figure 1). It is not cleared why the author selected only 128 patients in this study and Fig.1 is confusing to understand.

4. A small group of population main concern.

Out of 161, only 128 enrolled patients were included. Not clear about the missing population in this study. Is it due to side effects? 20 patients were

excluded because of their follow-up durations were less than 30 months (not clear about missing data during that time and follow up).

Excluded 28 of 128 patients who were undergoing ADT during follow-up (Figure 1)??

Two patients were lost to follow-up (not clear)??

A very small group of studies to compare: Of the remaining 98 patients, 60 (61.2%) patients had no biochemical recurrence and thus did not require any salvage treatment, and 38 (38.8%), patients received salvage treatment due to biochemical recurrence.

Conclusion: Large group of the population required for efficacy and safety of the treatment. Clarification required for lost to follow up.

5. Prostate-specific antigen: It is very important to know the PSA levels for prostate cancer patients. So, all patients had PSA levels measured three to six months for the first few years to check how well the HIFU and salvage treatment has worked. So, I would recommend a Table including prostate-specific antigen and histology after high intensity focused ultrasound.

6. MRI analysis: MRI may provide a more sensitive test than PSA, as it can detect disease not elevating PSA but causing a change in the MRI features of residual prostatic tissue. Second, when the disease is detected on MRI, imaging also provides the location of the disease and therefore has the added advantage of being able to guide biopsy and salvage therapy by MRI analysis and the image need to include in the picture.

7. The author showed that patients with the high-risk group had shorter BCR and salvage treatment-free survival than prostate cancer patients with a low and intermediate group (P<0.0001) (Figure 2 and Figure 3). To determine whether the high-risk group has independent prognostic value, univariate and multivariate analysis needs to be performed.

8. I would recommend providing a table of Kaplan-Meier analysis of metastasis-free survival in Figure 4. What is the probability? How do certain personal, behavioral or clinical characteristics chances of survival?

6. PLOS authors have the option to publish the peer review history of their article (what does this mean?). If published, this will include your full peer review and any attached files.

Reviewer #1: No

Reviewer #2: No

---

## [Author Response · Author response to Decision Letter 0]

9 Apr 2020

Dear editor and reviewers:

Thanks for the time and effort you spent assessing the previous version of the manuscript. 

For the previous table 3 (table 7 in the revised manuscript), the IIEF questionnaire was not done for most patients in that time frame, so we typed “N/A” for de novo erectile dysfunction. The column was deleted for irrelevance to the cohort of this article as reviewer’s suggestion. Other responses to the comments are as below and all of your suggestions are incorporated into revised manuscript. 

Reviewer #1: The authors provided medium term oncological outcomes of a single institution cohort of 128 men receiving whole gland HIFU for clinically localized prostate cancer.

1. I would specify that the study describes whole gland HIFU. Modify and introduce an abbreviation for whole gland HIFU throughout the manuscript.

 Thanks for this suggestion. Whole-gland HIFU may be abbreviated as WG-HIFU. It was revised in the manuscript. 

2. Abstract does not read well. For example, the first proposition does not fit neither with the background nor with the objective of the study. No information are given regarding the efficacy of salvage treatments. Please address this point.

 The abstract was revised according to your suggestion. Thanks!

3. Did all patients received a post whole gland HIFU biopsy? Please clarify

 Post-HIFU prostate biopsy was not routinely arranged. It might be arranged because of biochemical recurrence. It might also be arranged for patients not meeting criteria of biochemical recurrence but worry about continuous elevation of PSA. We will add this paragraph to manuscript (p5, line 122-125). 

4. A table (or a Kaplan meier in case the follow up biopsy protocol was not standardized) showing the results of the follow up biopsy would be useful.

 Tables 4 and 5 showing the results of post-HIFU biopsy was added to the paragraph. Thanks for the suggestion. 

5. Which were the criteria, if there were any, to choose the salvage treatment strategy? Please describe.

 Of patients with residual tumor noted, secondary HIFU will be arranged. Of patients with local lymph node metastasis suspected, salvage radiotherapy will be suggested. Salvage radiotherapy with half to one year of ADT is the standard strategy. We will add this paragraph to manuscript (p6, line 130-134). 

6. If the authors are keen on reporting data regarding the efficacy of salvage treatments is necessary to report also the outcomes of the population receiving any salvage treatment (as subgroup analysis and figures as supplementary)

 Thanks for the suggestion. Experience of post-HIFU salvage treatment is relatively scarce. Therefore, we think we could try to share our experience after years of HIFU use. However, only 38 patients received salvage HIFU or salvage radiation therapy in this study. Their outcome was shown in figure 1. We also added some explanation about salvage treatment. 

7. Since all men received a pre mpMRI, is it possible to add any information regarding the presence of visible lesions, PI-RADS score etc?

 Thanks for this suggestion. Actually, of all 128 patients enrolled, only 12 patients received pre-HIFU mpMRI and the rest 116 patients received pre-HIFU pelvic CT. Since the number of mpMRI is small in this series, we are afraid that we cannot provide any new information about this issue.

8. Regarding the functional outcomes, which represent secondary outcomes according to the study’s methods, only data regarding the study population should be reported. All information regarding not included cohort should be removed since not relevant.

 The irrelevant sentences and table were removed (p15). Thanks!

9. Unless preop IIEF is available, any data regarding post op IIEF should be removed since not reliable.

 Preoperative IIEF was added to the paragraph (p15, line 232-233). Thanks for the suggestion. 

10. Adverse events should be put in one single table and, if possible, compared for each procedure (provide a p value if possible)

 The table regarding adverse events was revised and shown in table 8. 

11. The first sentence of the discussion part is out of context and biased. Please change the beginning of this paragraph

 Thanks for the reminder. This paragragh was revised (p16, line 261-266). 

 

Reviewer #2: 

1. The major concern on this works is that it is supported the manuscript a small number of patients and a very short period in this manuscript.

 Thanks for the comment. This is a median 5-year follow-up study. We share our HIFU experience because there are only a few papers of HIFU treatment of prostate cancer.

2. Abstract-too much space is spent on descriptive data and not enough space on the main result of the study.

 The abstract was revised. Thanks for the suggestion!

3. The author mentions in materials and methods that 405 patients with prostate cancer have been treated with HIFU from December 2009 to February 2015. From December 2009 to February 2015, 161 patients who were newly diagnosed with prostate cancer were enrolled. In total, 128 patients were included (Figure 1). It is not cleared why the author selected only 128 patients in this study and Fig.1 is confusing to understand.

 We are sorry about the misunderstanding. In our institution, we performed the first HIFU in December 2009. From December 2009 to December 2019, when we submitted the original manuscript, there are 405 patients treated with HIFU. With the concern of adequate follow-up period, we only enrolled 161 patients treated with HIFU during December 2009 to February 2015.

4. A small group of population main concern.

Out of 161, only 128 enrolled patients were included. Not clear about the missing population in this study. Is it due to side effects? 20 patients were

excluded because of their follow-up durations were less than 30 months (not clear about missing data during that time and follow up).

Excluded 28 of 128 patients who were undergoing ADT during follow-up (Figure 1)??

Two patients were lost to follow-up (not clear)??

A very small group of studies to compare: Of the remaining 98 patients, 60 (61.2%) patients had no biochemical recurrence and thus did not require any salvage treatment, and 38 (38.8%), patients received salvage treatment due to biochemical recurrence.

Conclusion: Large group of the population required for efficacy and safety of the treatment. Clarification required for lost to follow up.

  Of the 128 patients, 28 patients received ADT after primary HIFU as a role of salvage treatment because of biochemical recurrence or slight elevation of PSA without biochemical recurrence. Ten of the 28 patients stopped ADT months later and no more PSA elevation occurred. Eleven of the 28 patients still receive ADT currently and PSA level remain stable. Seven of the 28 patients still receive ADT currently, but gradual elevation of PSA is noted. However, they refuse or hesitate about further salvage treatment such as radiation therapy or secondary HIFU. 

 Since 41 of 45 (91%) suffered from biochemical recurrence within 30 months during follow-up, therefore, we excluded patients whose follow-up duration is less than 30 months to avoid missing biochemical recurrence. 

5. Prostate-specific antigen: It is very important to know the PSA levels for prostate cancer patients. So, all patients had PSA levels measured three to six months for the first few years to check how well the HIFU and salvage treatment has worked. So, I would recommend a Table including prostate-specific antigen and histology after high intensity focused ultrasound.

 Thanks for the suggestion. The relationships between post-HIFU PSA and post-HIFU prostate biopsy was shown in table 6. 

6. MRI analysis: MRI may provide a more sensitive test than PSA, as it can detect disease not elevating PSA but causing a change in the MRI features of residual prostatic tissue. Second, when the disease is detected on MRI, imaging also provides the location of the disease and therefore has the added advantage of being able to guide biopsy and salvage therapy by MRI analysis and the image need to include in the picture.

 Really thanks for your suggestion. We have described this method in the discussion in the revised manuscript (p19, line 330-334). 

7. The author showed that patients with the high-risk group had shorter BCR and salvage treatment-free survival than prostate cancer patients with a low and intermediate group (P<0.0001) (Figure 2 and Figure 3). To determine whether the high-risk group has independent prognostic value, univariate and multivariate analysis needs to be performed.

 Univariate and multivariate analysis were performed and showed in table 3.

8. I would recommend providing a table of Kaplan-Meier analysis of metastasis-free survival in Figure 4. What is the probability? How do certain personal, behavioral or clinical characteristics chances of survival?

 Associated univariate and multivariate analysis was performed and showed in table 3.

---

## [Decision Letter · Decision Letter 1]

25 May 2020

PONE-D-20-00019R1

Cohort study of high-intensity focused ultrasound in the treatment of localised prostate cancer treatment: medium-term results from a single centre

PLOS ONE

Dear Dr. Chiang,

Thank you for submitting your manuscript to PLOS ONE. After careful consideration, we feel that it has merit but does not fully meet PLOS ONE’s publication criteria as it currently stands. Therefore, we invite you to submit a revised version of the manuscript that addresses the points raised during the review process.

ACADEMIC EDITOR: Please kindly address the minor issues from the reviewer. 

We look forward to receiving your revised manuscript.

Kind regards,

Jason Chia-Hsun Hsieh, M.D. Ph.D

Academic Editor

PLOS ONE

Additional Editor Comments (if provided):

Please kindly address the minor issues from the reviewer.

Reviewers' comments:

Reviewer's Responses to Questions

**Comments to the Author**

1. If the authors have adequately addressed your comments raised in a previous round of review and you feel that this manuscript is now acceptable for publication, you may indicate that here to bypass the “Comments to the Author” section, enter your conflict of interest statement in the “Confidential to Editor” section, and submit your "Accept" recommendation.

Reviewer #1: All comments have been addressed

Reviewer #2: (No Response)

2. Is the manuscript technically sound, and do the data support the conclusions?

Reviewer #1: Yes

Reviewer #2: (No Response)

3. Has the statistical analysis been performed appropriately and rigorously? 

Reviewer #1: Yes

Reviewer #2: (No Response)

4. Have the authors made all data underlying the findings in their manuscript fully available?

Reviewer #1: Yes

Reviewer #2: (No Response)

5. Is the manuscript presented in an intelligible fashion and written in standard English?

Reviewer #1: Yes

Reviewer #2: (No Response)

6. Review Comments to the Author

Reviewer #1: The authors did a pretty good job addressing all the comments.

Further minor comments:

1) Table 3: provide HRs for both univariate and multivariate in a clearer way so that the reader can understand the confounders. logHRs not required

2) Please clarify what endoscopic biopsy means

3) Table 5: provide both raw numbers and percentages for each figures

4) positive/negative of malignancies sounds pretty bad. Rewrite it.

5) Please be consistent with the use of the short WG-HIFU

Reviewer #2: (No Response)

7. PLOS authors have the option to publish the peer review history of their article (what does this mean?). If published, this will include your full peer review and any attached files.

Reviewer #1: No

Reviewer #2: No

---

## [Author Response · Author response to Decision Letter 1]

23 Jun 2020

Reviewer #1: 

Further minor comments:

1) Table 3: provide HRs for both univariate and multivariate in a clearer way so that the reader can understand the confounders. logHRs not required

 Table 3 was revised according to suggestion of the Biostatistics Center of our institution. 

2) Please clarify what endoscopic biopsy means

 Endoscopic biopsy means transurethral biopsy by monopolar loop. 

3) Table 5: provide both raw numbers and percentages for each figures

 In table 5, percentage was added for each figures. 

4) positive/negative of malignancies sounds pretty bad. Rewrite it.

 The terms were revised. 

5) Please be consistent with the use of the short WG-HIFU

 Whole manuscript was examined and the WG-HIFU is used consistently.

---

## [Editor Report · Decision Letter 2]

29 Jun 2020

Cohort study of high-intensity focused ultrasound in the treatment of localised prostate cancer treatment: medium-term results from a single centre

PONE-D-20-00019R2

Dear Dr. Chiang,

We’re pleased to inform you that your manuscript has been judged scientifically suitable for publication and will be formally accepted for publication once it meets all outstanding technical requirements.

Kind regards,

Jason Chia-Hsun Hsieh, M.D. Ph.D

Academic Editor

PLOS ONE

Additional Editor Comments (optional):

All the questions were answered adequately.
---

## [Editor Report · Acceptance letter]

8 Jul 2020

PONE-D-20-00019R2 

Cohort study of high-intensity focused ultrasound in the treatment of localised prostate cancer treatment: medium-term results from a single centre 

Dear Dr. Chiang:

I'm pleased to inform you that your manuscript has been deemed suitable for publication in PLOS ONE. Congratulations! Your manuscript is now with our production department. 

Kind regards, 

on behalf of

Dr. Jason Chia-Hsun Hsieh 

Academic Editor

PLOS ONE